# Research on the Preparation Process and Performance of a Wear-Resistant and Corrosion-Resistant Coating

**Xianbao Wang** [1] and **Mingdi Wang** [2,*]

1    School of Mechanical and Electric Engineering, Soochow University, Suzhou 215000, China; 18972666977@163.com
2    Director of Laser Intelligent Manufacturing Joint R&D Center, Soochow University, Suzhou 215000, China
\*    Correspondence: wangmingdi@suda.edu.cn

**Abstract:** In order to study the wear resistance and corrosion resistance of a composite material with a Fe316L substrate and Co-Cr-WC coating, Co-Cr alloy coatings with different mass fractions of WC (hard tungsten carbide) were prepared on a Fe316L substrate by laser cladding technology. The phase composition, microstructure and element distribution were analyzed by X-ray diffraction (XRD), scanning electron microscopy (SEM) and energy dispersive spectroscopy (EDS). The hardness of the samples was tested by a Vickers microhardness tester, the friction coefficient and wear amount of the samples were tested by a friction and wear tester, and the corrosion resistance of the samples was tested by an electrochemical corrosion workstation. The results showed that the macroscopic appearance of the coating surface was good without obvious cracks, and the microstructures were mostly equiaxed crystals, cellular crystals and dendrites. With the addition of WC, the structures near the particles became more refined and extended from the surface of the WC particles. When the WC content was 40%, defects such as fine cracks appeared in the coating. The average microhardness of the 30%WC-Co-Cr coating was 732.6 HV, which was 2.29 times that of the Fe316L matrix; the friction coefficient was 0.16, and the wear amount was $14.64 \times 10^{-6}$ mm$^3$ N$^{-1}$ m$^{-1}$, which were 42.1% and 44.47% of the matrix, respectively; the self-corrosion voltage of the cladding layer was 120 mV, and the self-corrosion current was $7.263 \times 10^{-4}$ A/cm$^2$, which were 30.3% and 7.62% of the substrate, respectively. The experimental results showed that the laser cladding Co-Cr-WC composite cladding layer could significantly improve the wear resistance and corrosion resistance of the Fe316L matrix under the optimal laser process parameters.

**Keywords:** laser cladding; composite alloy powder; wear resistance; corrosion resistance

## 1. Introduction

Laser cladding technology utilizes a composite alloy powder material with a high-performance ratio melted on the surface of the substrate by the high energy of the laser to improve the hardness, wear resistance, corrosion resistance and other properties of the surface of the parts [1]. The powder feeding method of laser cladding technology is divided into preset powder and synchronous powder feeding laser cladding. Preset powder cladding is when the powder is pre-arranged on the surface of the workpiece before the laser action forms the molten pool, and synchronous powder cladding is when the powder is simultaneously fed into the molten pool during the laser action process [2,3]. Both methods can prepare composite materials on the surface of the substrate. The alloy cladding layer, among the traditional surface engineering techniques of thermal spraying and electroplating, is considered to be the most efficient method for preparing coatings. Both of these methods are surface treatment processes used to improve the wear resistance, corrosion resistance, high-temperature resistance and heat insulation properties of the surface of equipment or components, and are widely used in the production of hydraulic supports, rolls, valves and other industrial products [4,5]. Due to the non-environmentally

friendly nature and low reliability of traditional surface engineering techniques, coating processes that can replace these traditional techniques have received extensive attention. Under the action of the laser beam, the alloy powder or ceramic powder and the surface of the substrate are rapidly heated and melted, and the powder alloy in the molten state is cooled to form a surface coating with a very low dilution rate and a metallurgical aggregate with the substrate material. The thermal influence on the surface of the part is very low [6]. During the solidification process of the cladding layer from the bottom to the top, non-equilibrium solidification structures such as saturated solid solution and crystallites will be formed. These crystals can greatly improve the comprehensive mechanical properties of the coating, so as to significantly improve the wear resistance, corrosion resistance, heat resistance, antioxidant and electrical properties of the substrate surface [7,8].

In the current research on laser cladding powder materials, self-fluxing alloy powders are mostly used in practical engineering applications [9]. Self-fluxing alloy powders generally include iron-based, nickel-based, and cobalt-based powders, to which elements such as silicon and boron are added to make the alloy powders have better self-fluxing properties. The Co-Cr-based alloy powder has good wear resistance and corrosion resistance [10]. In the preparation process of the cladding layer, Co and Cr react to form a solid solution, and other elements in the Co-based alloy react to form carbon and boron compounds dispersed in the Co-Cr solid solution, resulting in the comprehensive performance being greatly improved, so it is widely used in practical production. The surface hardness and wear resistance of the composite alloy cladding layer prepared by VP Biryukov using Co-Cr-based powder are significantly higher than those of the matrix babbitt steel [11]. The research results show that the laser cladding technology can not only be used to repair the surface of the workpiece, but also can be used to repair the inner wall of parts such as pump equipment, motor shafts, crank journals, and other parts such as sliding friction pairs and cladding, to increase their reliability and life. Li Yongquan, Jining, et al. studied the phase structure and friction and wear properties of Ni alloy coatings prepared by laser cladding on the surface of 45 steel [12]. They found that the coating and the substrate were well bonded, the two were metallurgically bonded, and the coating phase was mainly $Ni_3Cr_2$, NiTi, SiC, TiC and $\gamma$-Ni, etc. Under the same conditions, the wear quality of the coating was about 1/8 of the wear quality of the substrate, and the wear resistance of the coating was greatly improved compared with that of the substrate [13]. Wang Kaiming et al. prepared WC-Ni matrix composite cladding layers on the surface of Q235 steel substrate and studied the effect of different WC contents on the microstructure and wear resistance of the coating. They concluded that when the WC mass fraction is 20%, the cladding the layer dilution rate is the smallest [14]. With the increase of WC content, the grain refinement strengthening effect of the cladding layer is enhanced, and its wear resistance is improved. The phases of the coating containing WC particles are mainly $\gamma$-Ni, $M_7C_3$, $M_{23}C_6$, CrB, WC and $W_2C$ [15].

Based on the current knowledge in the literature, it is found that laser cladding technology has been gradually developed to strengthen the comprehensive performance of parts, but the related process research only discusses the influence of a single alloy material on the properties of the cladding layer [16]. Therefore, it is designed to mix Co-Cr alloy powder and hard phase WC with stronger hardness and corrosion resistance to form a composite powder material in order to enhance the hardness, wear-resistance and corrosion resistance of the Co-Cr-based cladding layer, respectively. Different mass fractions of WC hard phases were added to enhance the comprehensive mechanical properties of the composite powders. The WC hard phase has a high melting point, high hardness, and excellent wear resistance and corrosion resistance. It was used to strengthen the surface properties of the parts to study the microstructure, microscopic appearance, microhardness, resistance of laser cladding layers, abrasiveness and electrochemical corrosion, thereby improving the comprehensive surface properties of the cladding layer.

## 2. Powder Materials and Experimental Equipment

### 2.1. Powder Material

The Fe316L stainless steel plate was selected as the base material. The substrates were treated with sandpaper and acetone solution, respectively, then ultrasonically cleaned and dried with ethanol. The powder material used in the experiment is Co-Cr-based spherical powder with a particle size of 48–106 μm. The powder is a self-fluxing alloy powder, which has good self-fluxing properties because of B and Si elements, and has excellent characteristics such as deoxidation and slag formation, and prevention of oxidation of the cladding layer. The Co-Cr-based powder has excellent wear resistance, corrosion resistance and toughness, and its physical properties are still stable in high-temperature environments [17]. The chemical composition of the powder is shown in Table 1.

**Table 1.** Chemical composition of Co-Cr based powder (Quality Score %).

| Element | C | W | Ni | Fe | Cr | Si | Co |
|---------|-----|------|-----|-----|------|-----|--------|
| Unit | 2.2 | 12.7 | 0.2 | 0.3 | 29.8 | 1.3 | margin |
| MIN | 2.2 | 11.5 | 0 | 0 | 29.0 | 1.0 | margin |
| MAX | 2.5 | 13.5 | 2.0 | 3 | 32.0 | 1.5 | margin |

The prepared Co-Cr-based powder and the weighed WC ceramic hard phase with mass fractions of 10%, 20%, 30%, and 40% were put into a ball mill for mixing, and the ball milling speed was 45 r·min$^{-1}$ for 1 h. After mixing evenly, it was put in a drying cabinet for 1 h to ensure the drying of the mixed powder and marked as samples N2, N3, N4, and N5, respectively; the Co-Cr-based powder was marked as sample N1.

### 2.2. Laboratory Equipment

The laser cladding equipment used in this experiment was mainly comprised of a three-axis motion platform and its wide-spot laser cladding head, a fiber laser, a double-barrel powder feeder, a water cooler, and an inert gas delivery system [18]. Under the action of the control system, the laser cladding head can make an optimized motion trajectory according to the shape of the workpiece under the movement of the three-axis platform. The principle of the laser cladding processing is shown in Figure 1. At the same time, the control system opened the powder feeder and the inert gas conveying system to ensure that the powder ejected from the powder feeding port could accurately enter the molten pool generated by the laser beam, and melt and solidify on the surface of the substrate. The laser cladding head was fixed on a three-axis mobile platform, and the cladding head could make a corresponding movement trajectory according to the strengthening position required by the part through teaching programming or manual drawing [19].

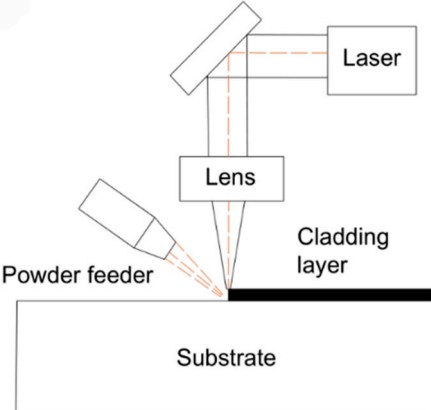

**Figure 1.** Laser cladding processing principle.

### 2.3. Characterization Methods

The sample was prepared by wire cutting, with a size of 15 mm × 15 mm × 2 mm, a tube pressure of 45 KV, a tube flow of 45 mA, and a diffraction angle range of 10° to 90° [20]. The selected target was Cu, and the scanning speed was 20°/min. The microscopic appearance and composition of the cladding layer were analyzed by a ZEISS EVO18 scanning electron microscope and its own energy dispersive spectrometer. The microhardness of the cladding layer was tested by an MH-5 digital Vickers microhardness tester with a load of 500 g and a loading time of 10 s. The friction properties of the cladding layer were tested by the ball-disk friction and wear testing machine [21]. The normal load was 9.8 N, the friction time was 45 min, the friction radius was 2 mm, and the friction disk speed was 1440 r/min. The abrasive material was $Si_3N_4$ high-hardness ceramic balls. The CHI electrochemical corrosion workstation was used to conduct the corrosion resistance test of the sample. The electrolytic cell included 3.5% NaCl solution, graphite auxiliary electrode and 3.5 mol/L AgCl electrode. The effective test area of the sample was 1 cm², and the range of the scanning motor selected for the polarization curve test was −0.75~−0.25 v (vs. SCE), and the scan rate was 2 mV/s. Although a rate of 2 mV/s was adopted in this stage of the experimentations, it is remarked that the potential scan rate has no substantial provided distortions in the polarization curves obtained [22–25]. Besides, no deleterious effect was verified when polarization parameters were obtained (e.g., corrosion current densities and potentials). However, it is worth noting that potential scan rate has an important role in minimizing the effects of distortion in Tafel slopes and corrosion current density analyses, as previously reported [22–25].

## 3. Results and Analysis

### 3.1. XRD Phase Analysis of Cladding Layer

Figure 2 shows the XRD pattern of the Co-Cr-based alloy cladding layer. By analyzing the diffraction peaks of the pattern, it can be seen that the phase composition of the Co-Cr alloy cladding layer includes $\gamma$-Co, $M_{23}C_6$, $Cr_7C_3$, $FeNi_3$, $Co_3W$, in which M mainly includes Ni, Cr, Fe elements, which is basically the same as the element composition in the alloy powder, but also contains $FeNi_3$, etc. This is because, during the laser cladding process, Fe316L in the molten pool penetrated into the molten pool through convection. In the cladding layer, some elements of Fe and Mn appear. Among them, $\gamma$-Co can not only enhance the hardness of the cladding layer, but also have wrapping and supporting effects on carbides such as $M_{23}C_6$ and $Cr_7C_3$. The crystal structures of the alloys are body-centered (bcc) and face-centered (fcc) cubic structures, respectively [26]. The analyte phase shows that the sample is a cobalt-chromium-iron-based cladding layer reinforced by various carbide hard phases. These main phases are determined. The comprehensive mechanical properties of the cladding layer are guaranteed, and the high hardness and high wear resistance of the cladding layer are guaranteed. In addition, due to the high content of Cr in the Co-Cr-based alloy powder, the diffraction peaks of carbides such as $M_{23}C_6$ and $Cr_7C_3$ are very high, indicating that their content in the cladding layer is large, and these carbides can ensure that the cladding layer has a higher hardness and wear resistance. When different mass fractions of WC hard phases were added to the Co-Cr alloy powder, carbide phases with higher hardness such as $CCo_2W_4$, $W_2C$ and WC appeared in the cladding layer. Because there are no elements such as C and Fe in the Co-Cr powder, the dissolved and diffused C elements of the WC particles form austenite by solid solution with elements such as Cr, Fe, and Co, and part of the austenite is rapidly cooled to form $M_{23}C_6$. When the mass fraction is 40%, a large amount of C makes the diffraction peaks of the $Cr_7C_3$ and $M_{23}C_6$ phases the most obvious. With the increase of WC content, more and more kinds of carbides are formed, and the diffraction peaks of the cladding layer become more obvious, indicating that the addition of WC can promote the production of carbides in the cladding layer, and the amount of these carbides can be increased. To sum up, it can be seen from the phase analysis that the hardness of the composite alloy cladding layer with the addition of the WC ceramic hard phase has been greatly improved, while

the hardness of the Co-Cr-40%WC cladding layer has reached its highest value, and its toughness also can be improved [27].

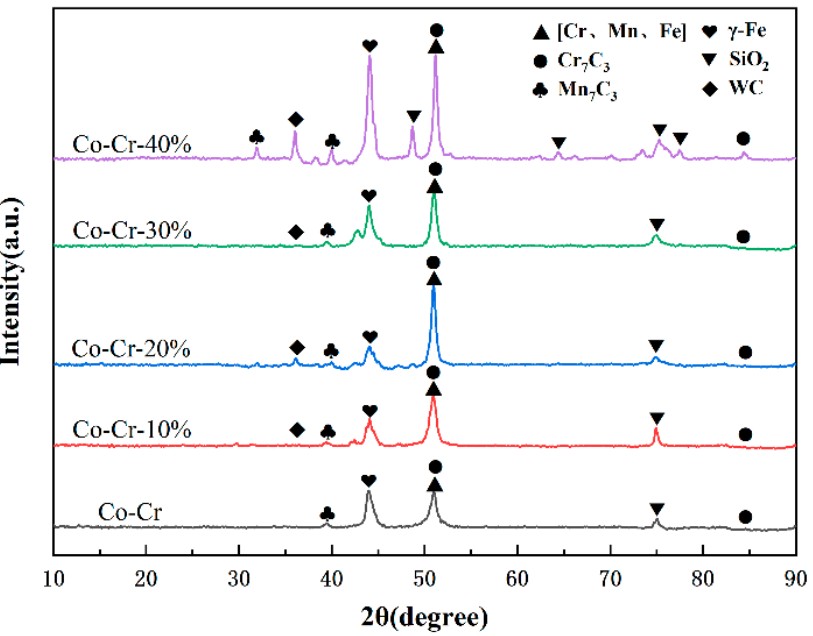

**Figure 2.** XRD pattern of Co-Cr composite cladding layer.

*3.2. Microstructure Analysis of Cladding Layer*

Figure 3 shows the metallographic microstructure of the junction, middle and top of the Co-Cr-based alloy powder cladding coating respectively. From Figure 3b, it can be seen that the cladding layer and the matrix are all bonded in a uniform bonding line, there are no obvious defects such as pores and cracks on the surface of the cladding layer, indicating that the laser cladding process parameters are suitable, and the Co-Cr alloy cladding layer has good adhesion to the Fe316 stainless steel substrate. Because when the heat input is too high due to the inappropriate laser power, scanning speed and powder feeding speed, the morphology of the bonding zone between the cladding layer and the substrate will be wavy due to the high temperature; when the heat input is too low, the powder feeding the powder sprayed by the nozzle into the spot cannot be completely melted, resulting in defects such as cracks and pores in the cladding layer. During the laser cladding process, the cladding powder flow in the cladding state cools first on the surface of the substrate, the temperature difference between the bottom and the top of the cladding layer is extremely large, and the temperature difference and the solidification speed form a large contrast [28]. The crystal growth rate is much faster than the nucleation rate. It can be seen from the bottom of the cladding layer in Figure 3g that the crystals extend to the interior of the cladding layer in the form of planar crystal growth at the junction with the substrate. With the progress of the cladding process, the matrix itself absorbs the huge amount of heat generated by the laser energy, and the temperature rises; the molten metal powder gradually solidifies, the temperature drops, the temperature difference gradually decreases, and the degree of microstructure grain refinement is much greater than that of the molten metal [29]. The part of the cladding layer close to the substrate is shown in Figure 3j; as shown in Figure 3h, it is the crystal state of the top of the Co-Cr alloy cladding layer. During the solidification process of this area, the molten state of the powder is still surrounded by high-temperature crystals that have just been formed, so the temperature difference between the two is not large, and its solidification reaches its maximum value. The crystals are rapidly cooled and formed before they grow. From the figure, we can see that there are many cellular and equiaxed crystals that extend in all directions and have smaller crystal sizes. This is also the reason why laser cladding can refine the microstructure of the cladding layer.

When the WC hard phase with different mass fractions was added, the growth state of dendrites changed [30]. When the heat input was too high due to inappropriate laser power, scanning speed and powder feeding speed, the high temperature between the cladding layer and the substrate caused the morphology of the bonding zone to be wavy; when the heat input is too low, the alloy powder cannot be completely melted, and metallurgical bonding occurs with the matrix during the solidification process, causing defects such as cracks and pores in different areas of the cladding layer. During the laser cladding process, the cladding powder flow in the cladding state cools first on the surface of the substrate, and the temperature difference between the bottom and the top of the cladding layer is extremely large. The cold theory shows that the temperature difference and the solidification speed have a large contrast, and the crystal growth speed in the cladding layer is much greater than the nucleation speed. As shown in Figure 3h, the central area of the Co-Cr-20%WC cladding layer has many WC particles distributed on the surface. The unmelted WC particles in the molten alloy powder have the lowest temperature and dendrites. Due to the hindered growth direction of the unmelted WC particles, the growth direction is no longer from the matrix to the top, but from the surface of the WC particles to the surrounding area, and many small, neatly arranged stacked cellular crystals appear, and the original needle-like dendrites and the columnar crystals become more refined, the volume of the interdendritic network becomes larger, and the overall structure becomes denser, as shown in the microstructure around the WC particles in Figure 3i [31]. When the WC mass fraction is high, as shown in Figure 3l, the content of unmelted WC particles is also higher, the grains of the cross-section of the cladding layer are more refined, and most of them are cellular crystals with smaller volumes. The structure is neatly stacked, dense and without obvious defects, so the comprehensive physical properties are also higher.

As shown in Figure 4, the SEM images of the cross-sectional joint area, the middle and the top area of the cladding layers of the samples N1, N3, and N4, respectively, are shown. From the micro-SEM image of the whole section of the cladding layer in the figure, it can be seen that the sizes of dendrites in different parts of the cladding layer of the sample are significantly different. It is because during the solidification process of the molten pool, the temperature gradient of different positions has different solidification rates, and the temperature of the molten pool decreases from the bonding area to the top of the cladding layer, as shown in Figure 4d,e,g. As shown in Figure 4h, the crystal state of the middle and top regions of the cladding layer is compared, so the dendrite matrix in different positions has different nucleation and growth rates, and the temperature in the top region changes most rapidly, so the grain size is smaller than that in the middle region [32]. Furthermore, the microstructures at the top of the samples are all cellular crystals, which are smaller in volume and denser than the needle-like dendrites that exist in large quantities in the middle and binding areas; as shown in Figure 4d, the sample (0% WC) microstructure is mainly composed of needle-like dendrites and columnar crystals and contains a small number of cellular crystals. When the WC hard phase with different mass fractions is gradually added, under the optimal process parameters, the morphology of WC particles in the cladding layer is mainly divided into two types, one is the slightly dissolved-diffusion type, in which the WC particles are relatively complete and the surface morphology is relatively clear, but the edges and corners are more dissolved and the original shape is basically maintained. The reason is that the heat generated by the laser beam is not enough to dissolve the WC particles at the bottom due to high melting point elements such as Cr and W. For the dissolution-diffusion type, the edges and corners of WC have basically been dissolved, resulting in more protruding needle-like structures. Dissolution and diffusion occur on the surface [33]. With the increase of WC content, the structure of the cladding layer becomes denser, as shown in Figure 4f, the micrograph of the cladding layer with 30%WC addition and the dendrites near the unmelted WC particles. The growth was hindered, and a large number of needle-like dendrites and columnar crystals without a WC cladding layer were gradually refined into cellular crystals, and the cellular crystals were smaller than the original grains, indicating that the unmelted WC particles were melting the microstructure

of the cladding, playing a role in grain refinement. In addition, at different positions of the cladding layer, dendrites and interdendritic networks will form different phases due to the inclusion of W, C and other elements, such as carbides of different elements, resulting in different sizes of organizational structures. Therefore, it can be observed that the structure of each part of the Co-Cr-30%WC composite cladding layer is the smallest and densest, and the comprehensive physical properties are optimized [34].

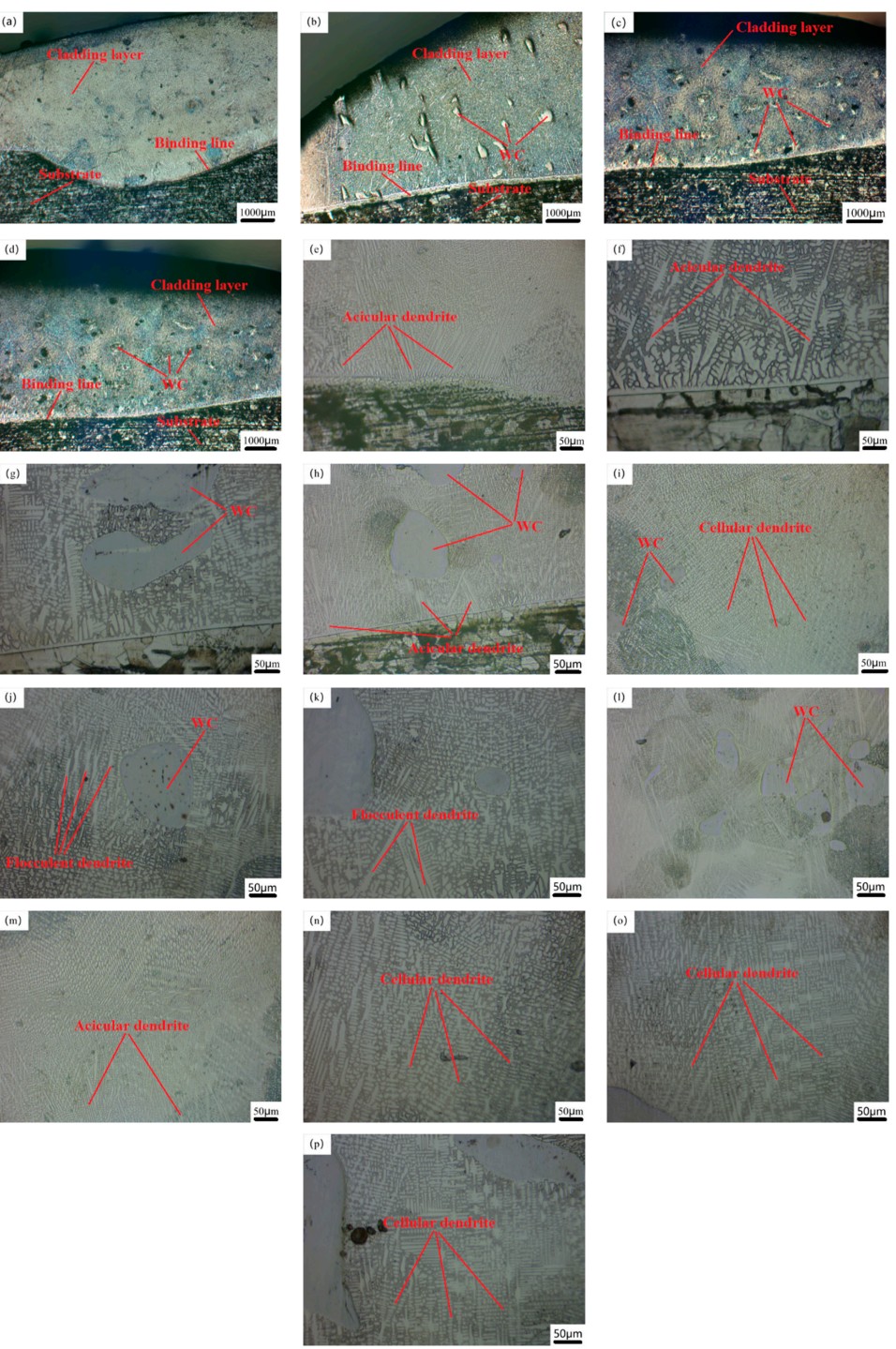

**Figure 3.** Metallographic diagram of Co-Cr-WC composite cladding layer. (**a**,**e**,**i**,**m**) Co-Cr; (**b**,**f**,**j**,**n**) Co-Cr-10%WC; (**c**,**g**,**k**,**o**) Co-Cr-30%WC; (**d**,**h**,**l**,**p**) Co-Cr-40%WC.

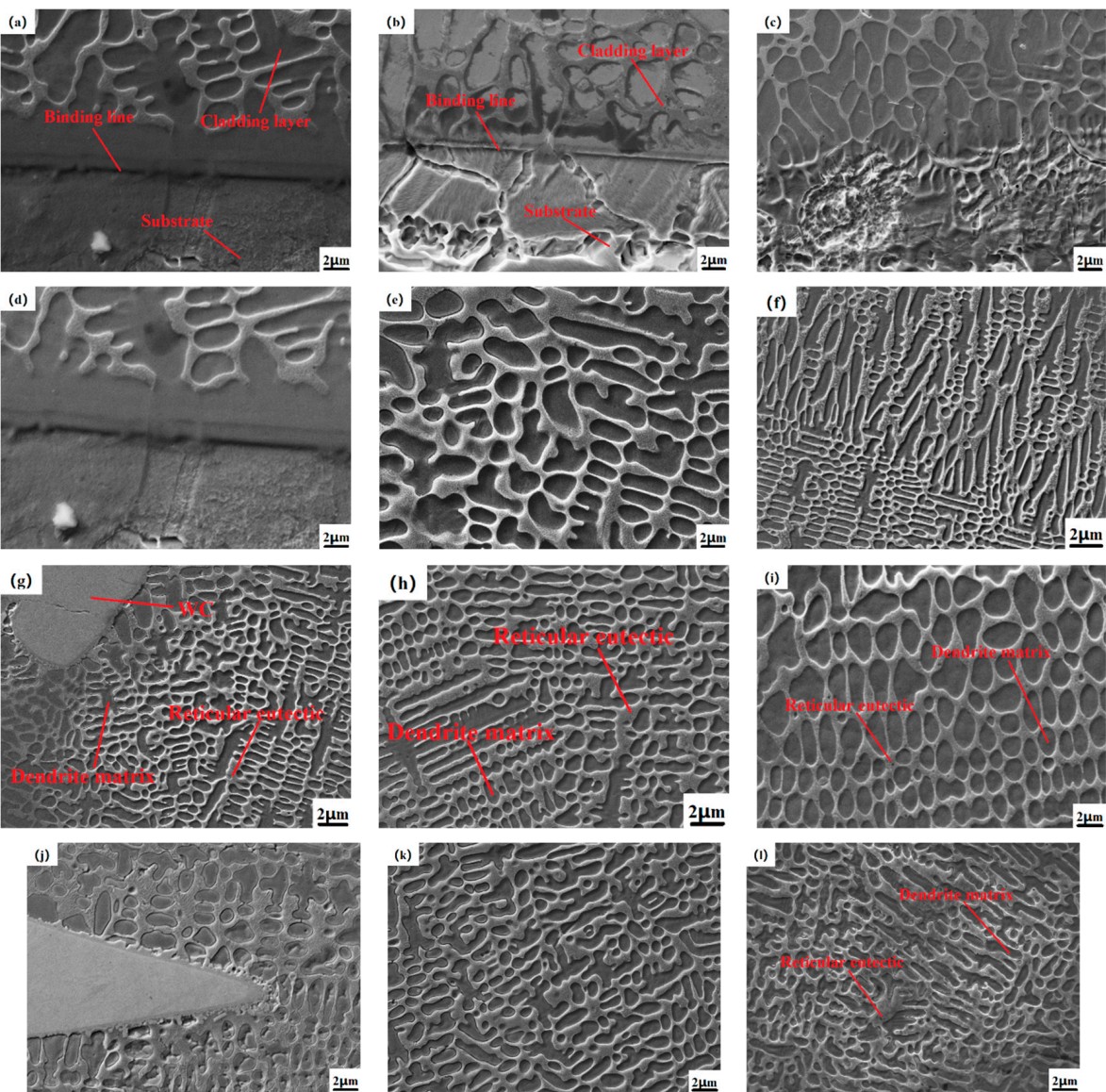

**Figure 4.** SEM image of Co-Cr-WC composite cladding layer. (**a,e,i**) Co-Cr; (**b,f,j**) Co-Cr-10%WC; (**c,g,k**) Co-Cr-30%WC; (**d,h,l**) Co-Cr-40%WC.

Figure 5 shows the micro-dissolution-diffusion WC particle diagram of the Co-Cr-30%WC cladding layer. When the composite cladding layer contains WC hard phases with different mass fractions, the microstructure of the WC-based cladding layer needs to be measured on its surface elements to analyze the changes in the elements of the cladding layer and the elements in the powder, as well as various elements in the dendrite and network structure. Figure 6a shows the element surface analysis of the Co-Cr-WC composite cladding layer, and Table 2 shows the element point analysis of Figure 5, and contains elements with smaller mass fractions such as Ni and Si. The interdendritic network eutectic structure contains a large amount of Cr elements, mainly in the form of $Cr_7C_3$. At the same time, the surface microstructure of the cladding layer is a columnar and flaky interdendritic structure, which is denser. These hard phases can greatly improve the melting point, hardness, wear and corrosion resistance of the coating. It can be found in the XRD pattern that the diffraction peaks are mainly in the $\gamma$-Co solid solution; this is because, in the process of solidification and forming of the alloy powder in the molten state, elements such as Ni and Cr combine with Fe elements to form an iron-containing solid solution, thereby making the alloy cladding. The microstructure of the layer is denser, and

its hardness, wear resistance and corrosion resistance are also greatly improved in terms of macroscopic properties. Co-Cr alloy powder originally contains a small amount of the Si element. In the process of laser cladding, the Si element is mainly used for deoxidation during the solidification of the alloy in the molten state and is evenly distributed in the intradendritic and interdendritic network's eutectic structure. When adding different mass fractions of WC hard phase, it can be seen in the SEM image that the WC particles in the cladding layer mainly include micro-dissolution-diffusion type and dissolution-diffusion type. The point-scanning EDS analysis at the four points of D all detected W and C elements, and the diffraction peak of the W element appeared on the diffraction pattern. This is because, in addition to the unmelted WC particles, the elements of the other melted WC diffused into the dendrites of the intradendritic and interdendritic cladding layers. The molten W element is mainly distributed in the crystal, and the W element mainly exists in the crystal as a Fe-W-C-Cr carbide, which gives the cladding layer a higher hardness and wear resistance. In Figure 5b, it can be seen that there are no cracks and melting pits on the surface of the unmelted WC particles, and the crystal growth is hindered by the particles, making the original Co and Cr elements more uniform. A large amount of Cr element in the microstructure crystal can improve the toughness of the material surface, and the carbides between grain boundaries such as $Cr_7C_3$, $M_{23}C_6$, $Co_3W$, $CCo_2W_4$, $W_2C$, and WC can improve the hardness and wear resistance of the phase, so the WC is hard. The increase of the phase content can comprehensively improve the comprehensive mechanical properties of the cladding layer [35].

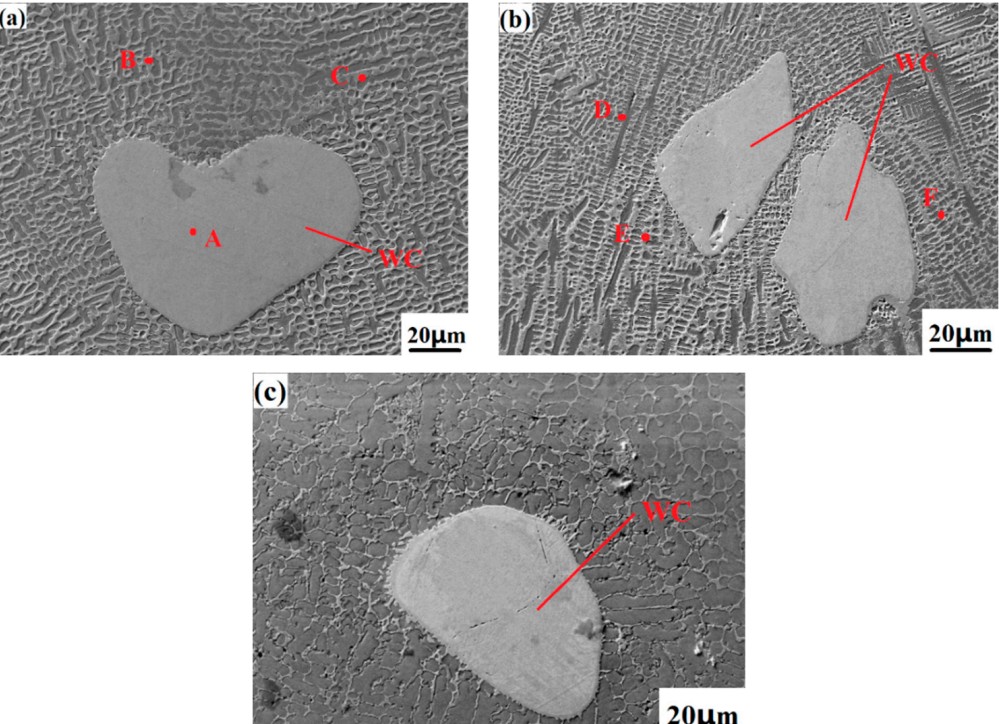

**Figure 5.** Microstructure image around WC particles in Co-Cr-30%WC cladding layer. (**a**) Slightly dissolving-diffusing WC particles; (**b**) Slightly dissolving-diffusing WC particles; (**c**) Slightly dissolving-diffusing WC particles.

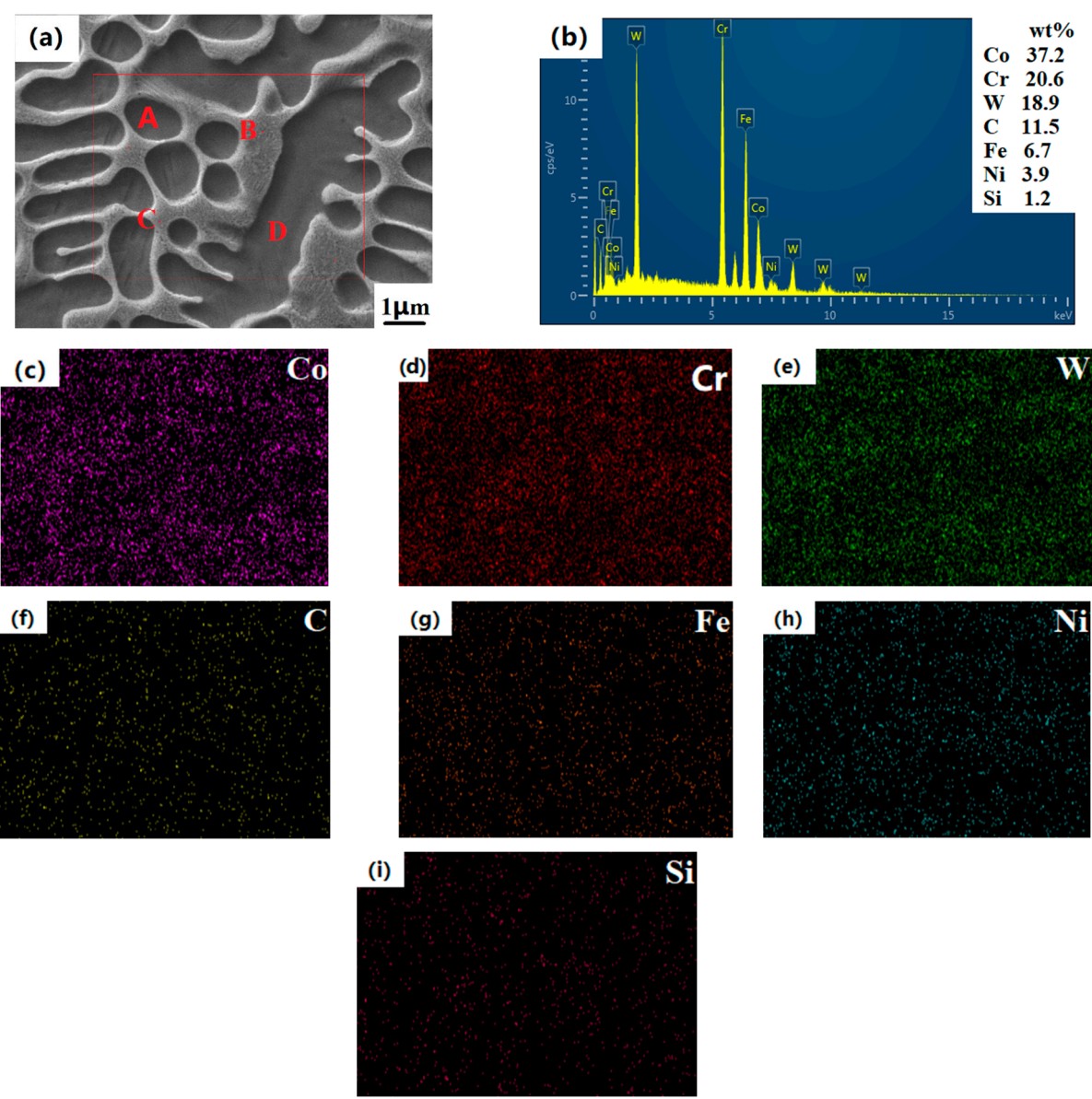

**Figure 6.** EDS surface scanning analysis of the middle area of the Co-Cr-30%WC cladding section. (**a**) Surface scanning area; (**b**) EDS surface total spectrum; (**c**) Co element distribution; (**d**) Cr element distribution; (**e**) W element distribution; (**f**) C element distribution; (**g**) Fe element distribution; (**h**) Ni element distribution; (**i**) Si element distribution.

**Table 2.** Analysis of EDS points in the section of Co-Cr-30%WC cladding layer (wt.%).

| Point | Co | Cr | W | Fe | Ni | C | Si |
|---|---|---|---|---|---|---|---|
| A | - | - | 86.5 | - | - | 13.5 | - |
| B | 26.3 | 23.6 | 35.1 | 5.2 | 4.4 | 4.2 | 1.2 |
| C | 32.9 | 19.6 | 29.9 | 6.3 | 3.6 | 6.1 | 1.6 |
| D | 23.9 | 25.2 | 36.4 | 4.7 | 3.4 | 4.9 | 1.5 |
| E | 28 | 18.6 | 34.5 | 6.1 | 4.8 | 5.5 | 2.5 |
| F | 22.5 | 23.4 | 33.7 | 5.7 | 5.3 | 6.5 | 2.9 |

### 3.3. Microhardness of Cladding Layer

Figure 7 shows the microhardness curves of different heights of the N1-N5 cladding layer section of the sample. The microhardness curve of the cladding layer is divided into three stages, which are the cladding layer, the heat-affected zone and the Fe316L matrix.

N1, N2, and N3 in the cladding layer and the heat-affected zone are first low and then high. The microhardness is 712.8HV0.5, 748.9HV0.5, and 749.8HV0.5, respectively. Among them, the microhardness of the surface of the cladding layer is the highest in the sample N5 with a mass fraction of WC of 40%, the average microhardness is 775.6HV0.5, and the average microhardness of the Fe316L matrix is 320.6HV0.5. The average microhardness of the cladding layers of 0%, 10%, 20%, and 30% are 552.6HV0.5, 649.8HV0.5, 694.3HV0.5, and 732.6HV0.5, respectively. The average measured microhardness of N1, N2, N3, N4 and N5 samples was 1.7, 2.0, 2.16, 2.28 and 2.42 times higher than that of the Fe316L matrix, respectively. With the addition of WC, the elements W and C generated by the dissolution-diffusion of WC particles interact with elements such as Co, Cr, and Fe to produce new hard phases such as $Cr_7C_3$, $M_{23}C_6$, $Co_3W$, $CCo_2W_4$, $W_2C$, and WC. The strength of the cladding layer is far greater than that of the pure Co-Cr-based cladding layer; in addition, the undissolved WC particles in the cladding layer can hinder the growth of columnar crystals and dendrites, which can refine the grains and make them denser and more uniform, which can be seen from the fact that with the increase of WC mass fraction, the microhardness curve of the cladding layer is smoother in the three parts of the cladding layer area, the heat-affected zone and the matrix material, and the fluctuation range is reduced [36]. It shows that the addition of WC makes the overall structure of the cladding layer more uniform and dense, and forms a good metallurgical bond with the matrix. In summary, the addition of WC ceramic hard phase can improve the microhardness of the cladding layer, make the microstructure more uniform, and make a smooth transition in the bonding area between the matrix and the cladding layer, reducing the hardness difference in each area.

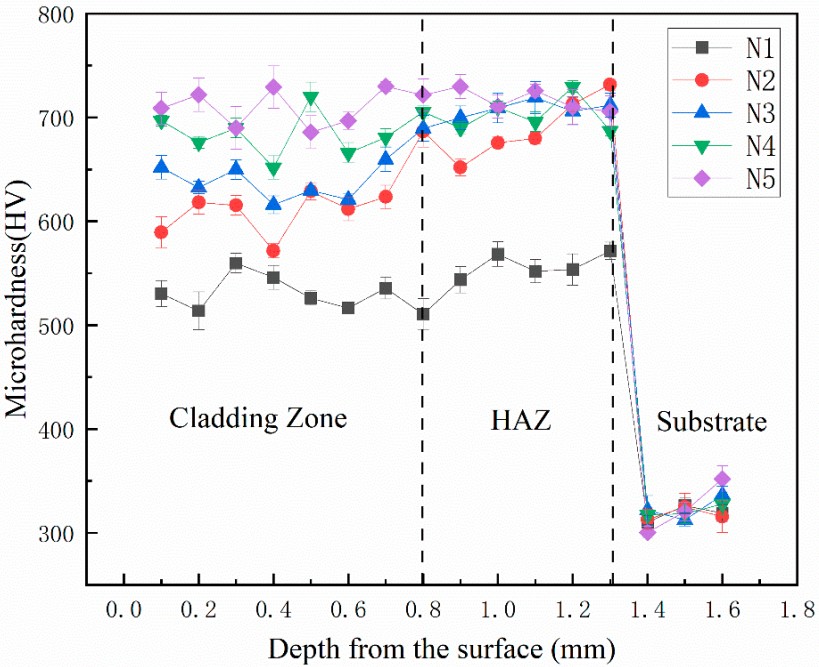

**Figure 7.** Microhardness of Co-Cr-WC composite cladding layer.

### 3.4. Friction and Wear Properties of Cladding

Figure 8a shows the friction coefficient of the Fe316L matrix and the Co-Cr-WC composite alloy cladding layer with different mass fractions of WC under the same test conditions. The friction coefficient curve of the sample is divided into the starting stage and the stable stage. At the beginning of the test, the grinding ball begins to contact the surface of the sample. The curve changes sharply up and down, so when the curve enters the stable stage, it will go through the stage of rising or falling in vain. When friction pits appear on the surface of the cladding layer, the curve tends to be stable. Under the condition of

25 N load and 45 min friction time, the average friction coefficient of Fe316L is 0.38, and the average friction coefficient of Co-Cr-WC composite cladding layer is 0.25, 0.23, 0.18, 0.16, and 0.15, respectively. It can be seen from the figure that the Fe316L matrix friction coefficient curve is at the top of the whole graph, and the friction coefficient is the largest; the friction coefficient curve of the Co-Cr-WC composite alloy cladding layer with different mass fractions increases with the increase of the WC mass fraction [37]. The coefficient decreases gradually, and the friction coefficient of the Co-Cr-40%WC cladding layer is the smallest, but the friction coefficient of the alloy cladding layer of the five samples has little difference. Therefore, with the continuous increase of the WC mass fraction, the microstructure of the Co-Cr-WC composite alloy cladding layer is more refined due to the addition of WC to the dendrite structure, and the W and C elements diffuse into the Co and Cr structure, and the reaction generates fused chromium carbides $Cr_7C_3$, $M_{23}C_6$, $Co_3W$, and $CCo_2W_4$ forms hard carbides that are widely and evenly distributed in the cladding layer, so the hardness and wear resistance of the cladding layer are high, meaning its microhardness is increasing. The average friction coefficient is also decreasing.

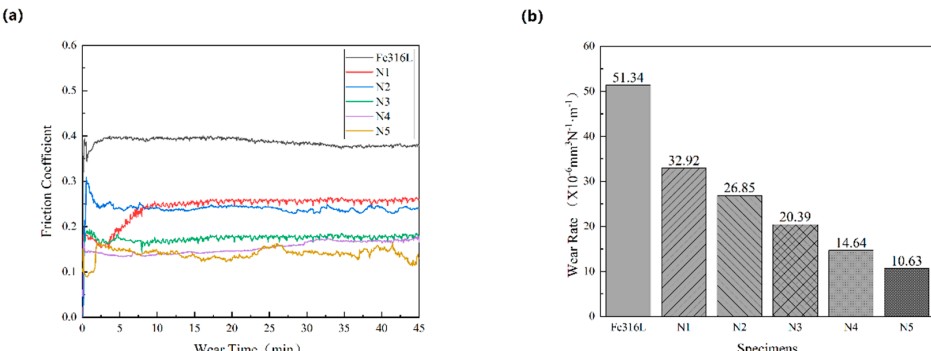

**Figure 8.** Friction and wear coefficient diagram and wear amount diagram of different cladding layers. (**a**) Friction and wear coefficient diagram of different cladding layers; (**b**) Comparison diagram of wear amount of different cladding layers.

Figure 8b shows the comparison of wear rates of Fe316L matrix and Co-Cr-WC composite alloy cladding layers with different mass fractions of WC under the same test conditions, and the wear rates are 51.34, 26.85, 19.63, 16.21, 11.02 and $5.21 \times 10^{-6}$ mm$^3$ N$^{-1}$ m$^{-1}$, respectively. The wear rate of the N6-N10 cladding layers of the sample is 52.3%, 38.24%, 31.57%, 21.46% and 10.15% of the base Fe316L, respectively. When the powder contains the WC cemented carbide phase, due to the presence of unmelted WC particles in the cladding layer, the growth of dendrites is hindered during the cooling process, and the original dendrites become small-grained cellular crystals; the elements W and C diffuse into the Co and Cr structures and react to form fused chromium carbides $Cr_7C_3$, $CCo_2W_4$, $W_2C$, and WC, thereby improving the hardness and wear resistance of the cladding layer. In addition, the Cr, Si, W and C in the Co-Cr-WC powder are weakly bonded between the graphite-like layers, and dislocation slip easily occurs along the basal plane direction. When the surface is loaded, the dislocation shift energy greatly increases, which increases the wear resistance of the cladding layer.

Figure 9 shows the surface wear morphologies of the Fe316L matrix and samples N1-N5 under the same conditions. The surface wear of the base Fe316L is relatively serious, the depressions are large, and a large amount of tissue falls off, and the pits left by the falling off of wear debris can be observed on the surface of the base, showing the tearing appearance of plastic deformation, as shown in Figure 9a. Figure 9b shows the wear profile of the Co-Cr alloy cladding layer. Compared with the wear profile of the substrate, there is less material falling off the surface, and the size and depth of the pits are also smaller. The hardness of the coating is greater than that of the substrate, and the degree of softening of the substrate by heat during the grinding process is low, so the surface material falls off less, and the abrasive particles formed by the fallen material will also be relatively

weak. When the cladding layer contains WC hard phases with different mass fractions, with the increase of WC content and the pit depth of the wear morphology of the N2, N3, N4, and N5 cladding layers decreases, and the surface material of the cladding layer falls off. It is also decreasing; an important reason for this is that as the proportion of WC increases, the hardness of the cladding layer is also increasing, and the wear resistance is also stronger. In addition, the surface of the Co-Cr alloy cladding layer in the interdendritic network structure phases is mainly chrome compounds such as $M_{23}C_6$ and $Cr_7C_3$. During the wear process, these chrome compounds act as lubricants between the grinding ball and the cladding layer, so that the friction coefficient of the friction pair is small and the friction is greatly reduced to protect the wear scar furrow from being further damaged by the lateral shear stress. When the WC hard phase is added, the W and C elements generated by the dissolution-diffusion of WC particles interact with Co and Cr to produce WC, $Cr_7C_3$ and other high hardness materials, and further enhance the wear resistance of the cladding layer. Figure 9a shows the overall wear appearance of the Co-Cr-10%WC composite alloy cladding layer. From the figure, it can be observed that the wear pits mainly show concave furrow-like wear marks. This is because the wear debris that fell off during the wear process is thrown out of the furrow by the grinding ball, and the residual wear debris in the furrow flows along the sliding track of the grinding ball, and acts as the abrasive particle between the friction pair between the grinding ball and the substrate. Under pressure, the surface of the substrate is extruded and sheared, so that the surface material undergoes directional plastic deformation, so the metal surface forms a pit shape with low middle and high sides. When the WC mass fraction increased to 40%, as shown in Figure 9d, the number of micro-dissolved-diffused WC particles on the surface of the cladding layer reached the maximum, and the wear pattern of the Co-Cr-WC composite cladding layer was the depth and size of the furrows which were significantly smaller than those of the pure Co-Cr-based alloy cladding layer without WC, indicating that the addition of WC hard phase enhances the hardness and strength of the cladding layer, and the wear is greatly reduced. To sum up, the dissolution-diffusion and slight dissolution-diffusion type WC hard particles can strengthen the Co-Cr cladding layer and improve its hardness and wear resistance.

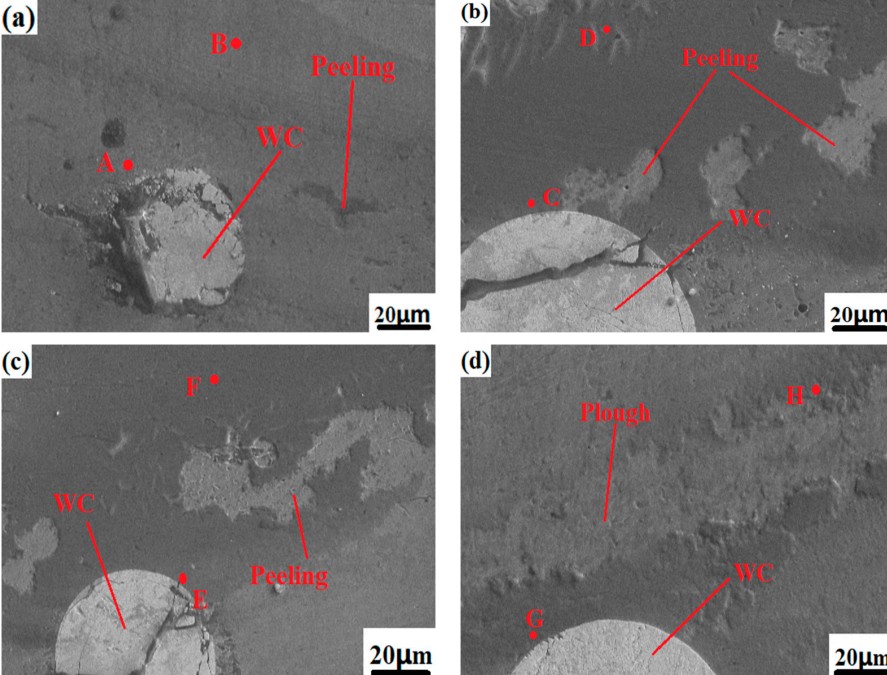

**Figure 9.** SEM images of surface wear of samples. (**a**) Co-Cr; (**b**) Co-Cr-10%WC; (**c**) Co-Cr-30%WC; (**d**) Co-Cr-40%WC.

Figure 10 shows the scanning diffraction pattern of the EDS surface after Co-Cr-30%WC wear, and Table 3 show the EDS element point analysis of the five points. A, B, C, D, E, and F in the table show that after 45 min of friction and wear, it is the same as the Co-Cr-WC composite alloy cladding layer with WC added. Oxygen elements were detected in the furrows of the wear scars, indicating that the friction and wear of the three samples all had oxidation wear. At the initial starting stage, when the cladding layer was in contact with the counter-grinding ball, the micro-morphology of the cladding layer was always the same. When the grinding ball was pressed and rubbed against the cladding layer, the contact part of the friction pair was slightly deformed, so the cladding layer material in the contact part fell off into granules, resulting in the melting of the surface material and the cladding layer being lost, and the wear mechanism is mainly adhesive wear. So the friction and wear of the cladding layer is actually a process of oxidative wear and mechanical wear. Due to the protective effect of the oxide film, the degree of oxidative wear on the sample is far less than that of adhesive wear. So in general, the oxide film can prevent the metal surface from sticking and play the role of protecting the friction pair. After adding different mass fractions of WC, the wear mechanism and EDS element detection content are similar to those of the Co-Cr-WC composite alloy cladding layer. Increasingly, the incompletely dissolved WC particles are evenly distributed on the surface of the cladding layer. During the wear test, the tissue material of the cladding layer is continuously peeled off, so that the WC particles with the highest hardness directly rub against the grinding ball, resulting in a large degree of wear. Therefore, after the mass fraction of the WC hard phase reaches a certain level, the oxidative wear and adhesive wear are greatly reduced to achieve the purpose of improving the wear resistance of the cladding layer. However, the Cr element existing in the Co-Cr cladding layer has the effect of lubrication and wear resistance. The chromium compounds generated by the interaction with elements such as W, C, and Fe are evenly distributed in the dendrite and interdendritic network structure. In the process, the friction coefficient of the cladding layer is reduced, and the degree of adhesive wear is further weakened to achieve the purpose of improving the wear resistance of the composite cladding layer.

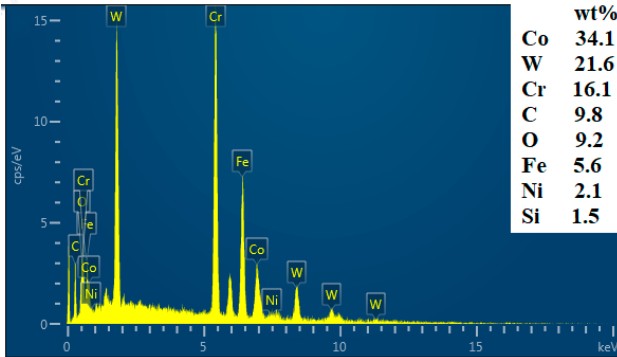

**Figure 10.** Scanning diffraction pattern of EDS surface after Co-Cr-30%WC wear.

**Table 3.** EDS point analysis of the worn surface of Co-Cr-30%WC cladding layer.

| Point | Co | Cr | W | O | Fe | Ni | C | Si |
|-------|------|------|------|------|-----|-----|------|-----|
| A | 33.8 | 23.9 | 15.6 | 8.6 | 5.8 | 5.2 | 5.6 | 1.5 |
| B | 31 | 26.7 | 16.9 | 7.9 | 4.5 | 6.8 | 4.4 | 1.8 |
| C | 24.5 | 21.9 | 24.9 | 9.5 | 6.7 | 3.5 | 6.5 | 2.5 |
| D | 19.1 | 27.5 | 26.4 | 10.1 | 4.8 | 5.7 | 4.5 | 1.9 |
| E | 22.2 | 20.1 | 28.6 | 6.9 | 6.8 | 4.7 | 7.5 | 3.2 |
| F | 23.3 | 18.1 | 30.4 | 7.0 | 5.4 | 5.6 | 7.8 | 2.4 |
| G | 24.8 | 15.9 | 29.6 | 8.6 | 3.5 | 5.3 | 10.5 | 1.8 |
| H | 14.4 | 21.3 | 28.9 | 9.1 | 8.1 | 4.9 | 11.8 | 1.5 |

Figure 11 shows the surface profile of the substrate and N1-N5 after friction and wear. Figure 11a,b shows the worn surface of Fe316L of the substrate. It can be seen from the figure that the contour of the substrate after wear is relatively rough, with a large number of particles and uneven places, and the wear scars are deep, indicating that its hardness is relatively high. In the process of wear, the tissue is worn and shed more, and the plastic deformation of the furrow is serious, so the surface morphology is irregular. Figure 11c,d shows the wear surface of the Co-Cr alloy cladding layer. Since its hardness is higher than that of the substrate, the degree of plastic deformation caused by extrusion during friction and wear is small, so the wear surface of the Co-Cr alloy cladding layer is higher. The marks are relatively shallow and the edges of the pits are well-defined. When the WC hard phase with different mass fractions was added, the surface roughness of the friction dent of the Co-Cr-based alloy cladding layer decreased significantly. The lower the degree of plastic deformation of the cladding layer, the stronger its ability to resist wear damage, and the surface integrity of the cladding layer becomes better under the same wear conditions. When there is no WC in the cladding layer, the wear type of the cladding layer is mainly for adhesive wear and oxidative wear, but when there are slightly dissolved-diffusion WC particles in the cladding layer due to the high hardness of the WC particles, they can directly contact the counter-grinding balls, thereby reducing the wear degree of the bottom and making the pits and furrows; the edge appears smooth. On the other hand, when the degree of wear decreases, less wear debris is generated during the friction process, and the degree of wear to the pits is gradually reduced, so with the increase of WC mass fractions N6, N7, and N8, the worn surface roughness of N9 and N10 cladding layers will also gradually decrease.

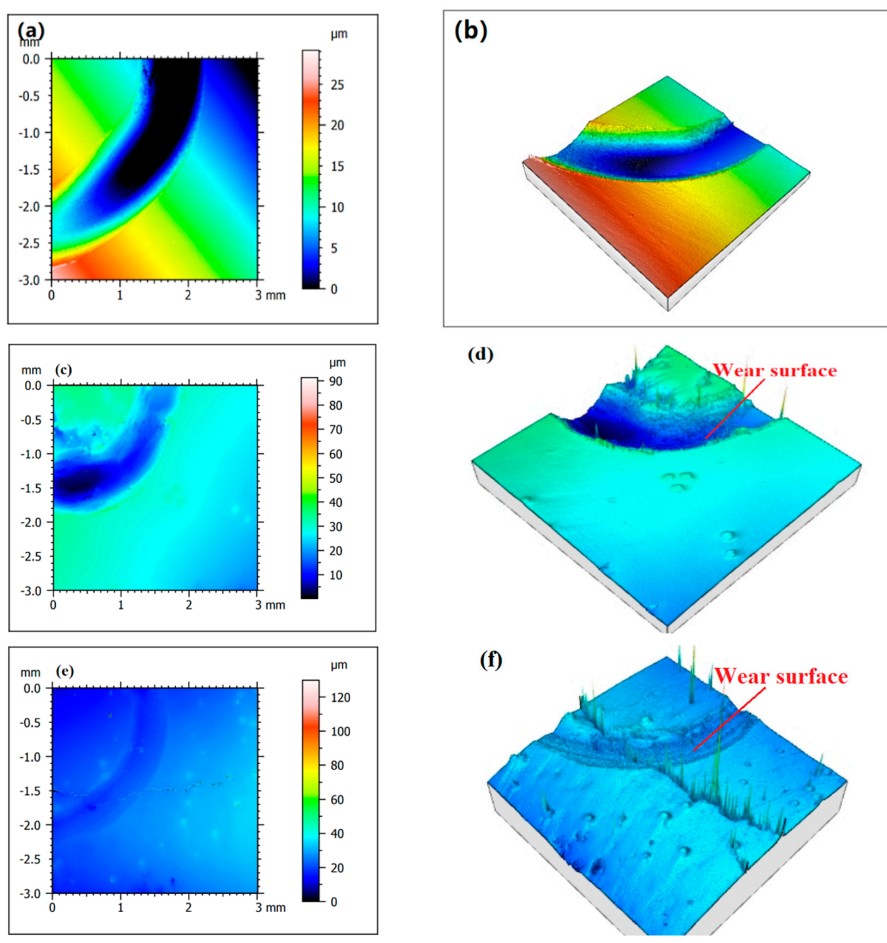

**Figure 11.** *Cont.*

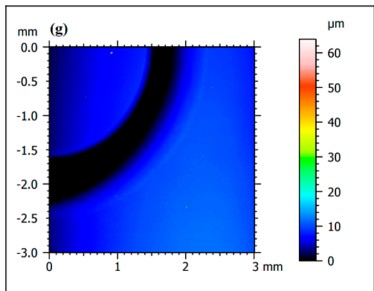
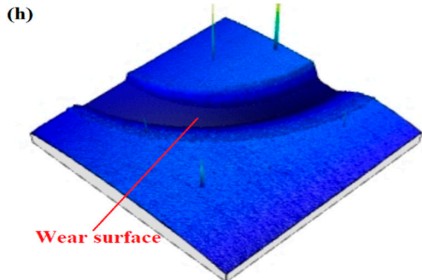

**Figure 11.** Profiles of worn surfaces of different cladding layers. (**a**,**b**) Fe316L; (**c**,**d**)Co-Cr; (**e**,**f**) Co-Cr-20%WC; (**g**,**h**) Co-Cr-40%WC.

### 3.5. Corrosion Resistance of Cladding

Figure 12 shows the potentiodynamic polarization curves of the Fe316L matrix and samples N1–N5 in a 3.5%NaCl solution. It can be seen from the figure that the corrosion current density of Fe316L increases significantly on the right side of the curve as the potential rises, indicating that the Fe element rapidly dissolves and iron ions are generated, and the dissolution rate of the material surface is greater than the formation rate of the passivation film, so the surface cannot form an effective passivation film. Corrosion electric density and self-corrosion potential are important parameters for evaluating the corrosion resistance of materials. The smaller the corrosion current density, the greater the self-corrosion potential, and the better the corrosion resistance of the material. Compared with Fe316L, the self-corrosion potential of the Co-Cr alloy cladding layer of sample N1 is significantly positive, the corrosion current density is $3.146 \times 10^{-3}$ A/cm$^2$, and the corrosion resistance is greatly improved. This is because the Cr content in the Co-Cr alloy powder reaches 29.8%. When the Cr content on the surface of the sample exceeds 12%, a dense passivation film can be formed on the surface of the sample during the test, which prevents the NaCl solution from diffusing into the interior and reduces the concentration of the surface of the sample and the cross-section of the solution improves the corrosion resistance of the cladding layer. Furthermore, under the optimal laser cladding parameters, the cladding layer has a dense structure and no obvious defects such as pores and cracks, so the corrosion resistance of the Co-Cr alloy cladding layer is much greater than that of the Fe316L matrix.

When different mass fractions of the WC hard phase were added, the self-corrosion voltages of samples N2–N5 gradually shifted positively, which were 280 mV, 243 mV, 120 mV, and 71 mV, respectively. As shown in Table 4, the self-corrosion currents are $1.592 \times 10^{-3}$ A/cm$^2$, $9.832 \times 10^{-4}$ A/cm$^2$, $7.263 \times 10^{-4}$ A/cm$^2$, and $4.263 \times 10^{-4}$ A/cm$^2$, respectively. The self-corrosion potential moves positively, the corrosion current density is significantly reduced, and the corrosion resistance is greatly improved. In the anode section of the curve, there is a passivation area in the process of increasing the corrosion current density, and the curve tends to be stable, indicating that a passivation film is formed on the surface of the cladding layer, which weakens the corrosion of the electrolyte and enhances the corrosion resistance of the material. When the mass fraction of the WC hard phase is 40%, the corrosion current density is the smallest, because the potential of the WC itself is much higher than that of the alloy cladding layer. It acts as a passivation film. Moreover, the microstructure of the cladding layer after adding WC is more uniform and dense, and the unmelted WC particles can well hinder the extension of the grains and play the role of refining the grains, thereby improving the corrosion resistance of the cladding layer.

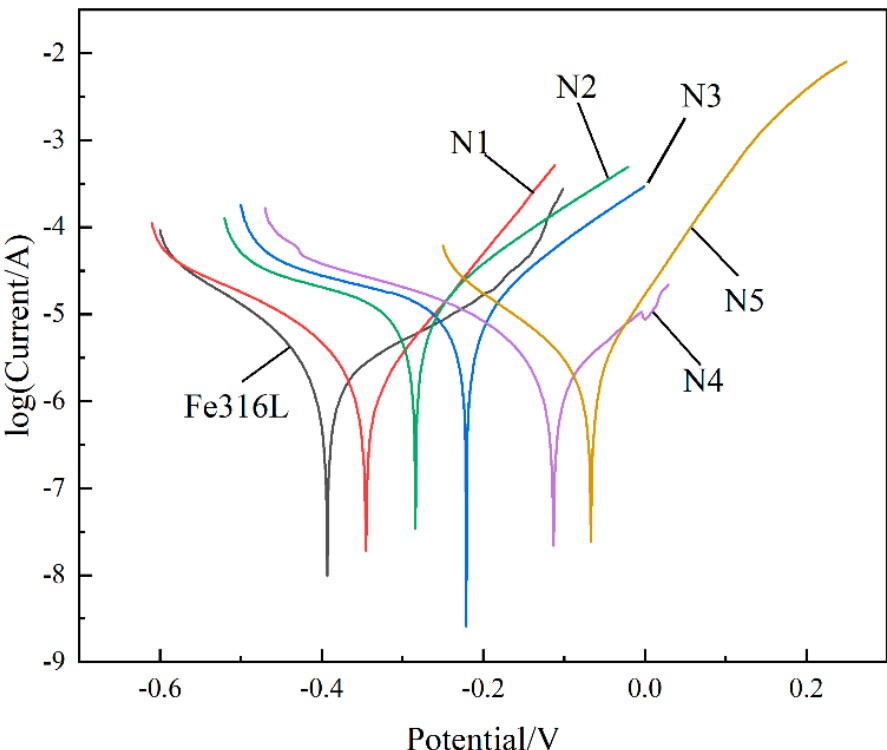

**Figure 12.** Potentiodynamic polarization curves of different cladding layers.

**Table 4.** Corrosion current density comparison of different cladding layers.

| Sample | $E_{corr}$/mV | $J_{corr}$/(um·cm$^{-2}$) |
|---|---|---|
| Fe316L | 396 mV | $9.534 \times 10^{-3}$ A/cm$^2$ |
| Co-Cr | 351 mV | $3.146 \times 10^{-3}$ A/cm$^2$ |
| Co-Cr-10%WC | 280 mV | $1.592 \times 10^{-3}$ A/cm$^2$ |
| Co-Cr-20%WC | 243 mV | $9.832 \times 10^{-4}$ A/cm$^2$ |
| Co-Cr-30%WC | 120 mV | $7.263 \times 10^{-4}$ A/cm$^2$ |
| Co-Cr-40%WC | 71 mV | $4.263 \times 10^{-4}$ A/cm$^2$ |

Figure 13 shows the appearance of the Co-Cr-WC composite alloy cladding layer with different mass fractions of WC after electrochemical corrosion, and Table 5 shows the corrosion of the Co-Cr-WC composite alloy cladding layer EDS point scan analysis. After the sample is electrochemically tested, it needs to be soaked in distilled water, rinsed with alcohol, and rinsed with acetone in order to remove the oxide film on the surface of the sample and the surface attachments after corrosion. It can be seen from the figure that the five samples have different degrees of corrosion. With the increase of the WC mass fraction, the corrosion resistance of the composite cladding layer is getting stronger and stronger. Among them, the Co-Cr-40%WC composite cladding layer has the strongest corrosion resistance, and the corrosion surface basically has no defects such as material peeling and corrosion pits. Different from the Ni-WC composite cladding layer, the Co-Cr-WC composite cladding layer contains more than 12% Cr, and colorless and transparent chromium-containing oxide will spontaneously form on the surface of the cladding layer. This layer of film can isolate the cladding layer from the medium and reduce the corrosion rate of the cladding layer. Therefore, it can be seen that the Co-Cr-WC composite cladding layer has stronger corrosion resistance than the Ni-WC composite cladding layer. Thanks to the Cr and microstructure of the Co-Cr-WC cladding layer, the compactness of the cladding layer can enhance the corrosion resistance of the cladding layer, and its passivation range is also the largest. There are dendrites and columnar crystals extending from the particle surface around the WC itself. Under the corrosion of the electrolyte, these crystal grains

are dissolved to different degrees, but the WC particles themselves do not appear slightly dissolved, indicating that the WC hard phase has an anti-corrosion effect. In addition, due to the addition of WC, the overall microstructure of the cladding layer is transformed into a phase with higher strength such as carbide and chrome, and the grains are more uniform and dense, so that it is difficult for the electrolyte to enter the interior of the sample and improve the corrosion resistance of the cladding layer. As shown in Figure 13c, the surface of the Co-Cr-40%WC composite cladding layer has almost no traces of corrosion after electrochemical corrosion, and no obvious corrosion pits are found on the surface of the cladding layer. Under the SEM scanning electron microscope, WC particles and their surrounding grains are basically complete, and the interdendritic network structure is still evenly distributed on the surface of the cladding layer, which can well prevent the electrolyte from entering the cladding layer and prevent the interior of the cladding layer's corrosion from being continued by the electrolyte; as such, this layer has the highest corrosion resistance.

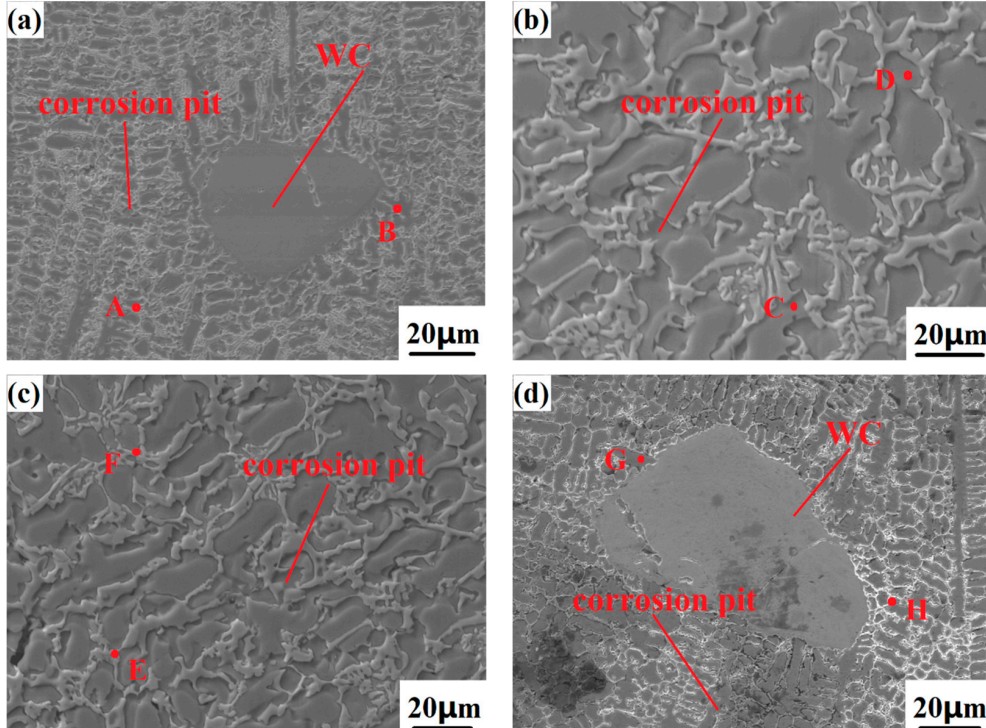

**Figure 13.** SEM surface corrosion morphologies of Co-Cr-WC composite alloy cladding layers with different mass fractions of WC. (**a**) Co-Cr-10%WC; (**b**) Co-Cr-20%WC; (**c**) Co-Cr-30%WC; (**d**) Co-Cr-40% WC.

**Table 5.** EDS point scanning analysis of Co-Cr-WC composite alloy cladding layer after corrosion.

| Point | Co | Cr | W | Fe | Ni | C | Si |
|---|---|---|---|---|---|---|---|
| A | 43.8 | 25.9 | 14.8 | 5.5 | 3.6 | 5.2 | 1.2 |
| B | 47.8 | 24.5 | 17.5 | 3.2 | 2.9 | 3.2 | 0.9 |
| C | 30.8 | 22.9 | 26.5 | 7.2 | 4.1 | 6.3 | 2.2 |
| D | 28.4 | 28.1 | 27.8 | 5.1 | 3.5 | 5.1 | 2.0 |
| E | 21 | 25.6 | 33.6 | 6.1 | 5.6 | 6.5 | 1.6 |
| F | 18.9 | 19.9 | 36.5 | 8.2 | 7.2 | 6.9 | 2.4 |
| G | 19.9 | 25.7 | 35.2 | 4.6 | 5.3 | 7.8 | 1.5 |
| H | 23.2 | 21.5 | 32.5 | 6.7 | 4.9 | 9.6 | 1.6 |

## 4. Conclusions

(1) The Co-Cr alloy cladding layer was prepared on the surface of Fe316L by laser cladding technology, and WC hard phases with different mass fractions were added to the Co-Cr alloy powder. Under the optimal laser cladding parameters, the main phases of the Co-Cr-based alloy cladding layer are $\gamma$-Co, $M_{23}C_6$, $Cr_7C_3$, $FeNi_3$, $Co_3W$, and the microstructure and crystallinity of the Ni-based and Co-Cr-based alloy cladding layers. The grains mainly include dendrites, cell-like crystals, strip-like crystals and interdendritic network structures. When the WC ceramic hard phase is added, the $Co_3W$, $CCo_2W_4$, $W_2C$, WC and other phases with higher hardnesses appear in the Co-Cr-WC composite cladding layer. The growth of dendrites near the unmelted WC particles is hindered, and the original large number of needle-like dendrites and columnar crystals are gradually refined into cellular crystals. The WC particles play a role in grain refinement in the microstructure of the cladding layer.

(2) The microhardness of the cladding layer increases with the increase of WC content. The maximum microhardness of the 30%WC-Co-Cr alloy cladding layer is 732.6HV0.5, which is 2.28 times that of the matrix; the minimum friction coefficient is 0.16, which is 42.11% of the matrix, and the wear amount is $14.64 \times 10^{-6}$ mm$^3$ N$^{-1}$ m$^{-1}$, which is 44.47% of the matrix. The profile of the cladding layer after friction and wear is relatively smooth, the furrow is relatively regular, and there is no obvious deformation; the wear mechanism is mainly adhesive wear.

(3) In the electrochemical corrosion test, the corrosion resistance of the Co-Cr-30%WC alloy cladding layer was better, the self-corrosion voltage was 276 mv positive relative to the substrate, and the self-corrosion current density was $7.263 \times 10^{-4}$ A/cm$^2$, which was the substrate of 7.62%. The Cr element and carbides on the surface of the cladding layer play a role similar to the passivation film in the electrolyte, and the microstructure was more uniform and dense, thereby improving the corrosion resistance of the cladding layer.

**Author Contributions:** Conceptualization, M.W. and X.W.; methodology, M.W.; software, X.W.; validation, X.W.; resources, M.W.; data curation, X.W.; writing—original draft preparation, X.W.; writing—review and editing, X.W.; project administration, M.W.; funding acquisition, M.W. All authors have read and agreed to the published version of the manuscript.

**Funding:** This research was funded by Suzhou key core technology research and development projects(Grant No. SGC2021010), Suzhou Gusu Technology Entrepreneurship Angel Program(Grant No. CYTS2020094).

**Institutional Review Board Statement:** Not applicable.

**Informed Consent Statement:** Not applicable.

**Data Availability Statement:** Not applicable.

**Conflicts of Interest:** The authors declare no conflict of interest.

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
