# Peer review of "Research on the Preparation Process and Performance of a Wear-Resistant and Corrosion-Resistant Coating"

_crystals, doi:10.3390/cryst12050591_

Round 1

Reviewer 1 Report

The paper was written in very poor quality with too many grammatical errors, which prevents readers from understanding what authors are trying to say. Without looking into the technical content, I have to recommend against the publication of this work. See some comments below, mainly for the Foreword section. I would suggest conducting a thorough check and revision on grammar, before its form is ready for submission to a journal.

  1. Abstract: “X-ray diffraction ( The microstructure and element distribution were an-10 alyzed by XRD)” Please make those two separate sentences instead of using “()”. Also, please separate the characterization techniques (XRD SEM EDX) with the mechanical property tests (hardness, wear resistance) into two sentences.
  2. Abstract: Please talk about the material details at the beginning, including the base materials as well as the coating materials. It’s until the end of the abstract when I realized the base material is 316L.
  3. Abstract: “the friction coefficient is 0.16, and the wear amount is 14.64 X10-6mm3N-20 1â–ªm-1, respectively 42.1% of the matrix and 44.47%;” Please correct the sentence. Also please use the superscript for the order and the units. 
  4. Foreword: “Laser cladding technology is a composite alloy powder material” should be “Laser cladding technology utilizes a composite alloy powder material”
  5. Foreword: “Alloy cladding layer[1-2].” Please correct this.
  6. Foreword: “Ni3Cr2, M7C3, M23C6, CrB, W2C, and etc.” Please use subscripts for the numbers in the chemical formula. Please check through the whole manuscript.
  7. Foreword: “Based on the research status at home and abroad” What do authors mean by “at home and abroad”?
  8. Foreword: “strengthen the surface strength” Please correct the sentence.
  9. Foreword: “the related 84 process research only stays in the process research stage” This sentence is not meaningful.
  10. Foreword: “stronger hardness and corrosion resistance” Please correct the phrase.
  11. Foreword: “Polish the substrate with 100#,200#300#,and400#sandpaper respectively, clean the oil stains on the surface of the substrate with acetone solution, then ultrasonically clean and dry with ethanol.” This is not a complete sentence.

Reviewer 2 Report

I can't review the article's merit before the authors make the corrections to the main manuscript text. Many sentences need general rewriting sense, grama, meaning. The authors should start from the title.

I also see a lot of general mistakes like the first sentence of the abstract the authors study a composite, not an alloy coating. In the second sentence, the XRD is not a method to analyze microstructure and element distribution etc.  

Reviewer 3 Report

Please explain the abbreviation of WC when using it for the first time (hard tungsten carbide).

Please edit carefully lines 52-54.

Insert in here also the impact of crystals on elastic properties. Is it add any drawback?

section 4 Capital letter

On line 69, replace etc with "at al"

Replace at home with "in our country"

Editing on line 90. Point has to be comma. Capital letter is needed on Section 2.1 and all subsections.

Complete this sentence "It can be seen at the bottom of the 190 cladding layer in Fig. XXXXXX"

Line 233 ; please complete the sentence: This is  because during the solidification process of the molten pool, different positions XXXXXXXXXXXX

Line 237 Fig XXX????

Line 444Remove And and begin with capital letter.

Table 4. please check and add other important parameters  calculated from tafel plots: R for instance...

Please reformulate: Among them, Co-Cr-40%WC composite cladding The corrosion resistance of the layer is the strongest, and there is basically no material peeling off the  corroded surface, and it is difficult to observe defects such as corrosion pits.

Figure 11, needs to be revised N1-5 has no meaning to correctly identify the samples. The text shoud be easily understandable.

I suggest to add a table to clearly identify the N1-N5 samples. The text after Table 1 is not enough for your description to be aqurate.

Comments regarding Tafel plots have to be in more detail presented. What about anodic and cathodic slope?

References are up-to-date.

The manuscript needs carefully editing all over the manusctript, but the approach and scientifical investigations are well done.

Reviewer 4 Report

REPORTS ON: crystals-1673333

The aim proposed is very interesting and novelty is provided. Before its final publication, certain weaknesses should be solved and a MAJOR REVISION is indicated, as follow:

  1. In the Abstract, a simple past tense should be used. At least three distinct tenses are used. Please, revise it.
  2. Also, into Abstract, there are exponent numbers that should be verified/revised. This also should be provided throughout the proposed manuscript.
  3. The section 2 and their subsections should be revised in order to include error ranges for a great number of physical elements and dimensions.
  4. Also, into subsections 2 the reproducibility is rather and poorly explained.
  5. Between lines 141/142, both work electrode and counter electrodes areas should be indicated. Their error ranges should also be indicated.
  6. Considering the scan rate (2 mV/s) written in line 142, the follow sentence and references should be included.

“Although a 2 mV/s is adopted in this stage of the experimentations, it is remarked that potential scan rate has no substantial provided distortions in the polarization curves obtained [AA-DD]. Besides, no deleterious effect is verified when polarization parameters are obtained (e.g. corrosion current densities and potentials). However, it is worth noted that potential scan rate has an important role in order to minimize the effects of distortion in Tafel slopes and corrosion current density analyses, as previously reported [AA-DD].”

[AA] Osório W.R., Freitas E.S., Garcia A, EIS and potentiodynamic polarization studies on immiscible monotectic Al–In alloys. Electrochimica Acta. 2013, 102: 436–445.

[BB] Osório W.R., Peixoto L.C, Moutinho D.J., Gomes L.G., Ferreira I.L., Garcia A. Corrosion resistance of directionally solidified Al–6Cu–1Si and Al–8Cu–3Si alloys castings. Mater. Design 32 (2011) 3832-3837.

[CC] Zhang X.L., Jiang Zh.H., Yao Zh.P, Song Y., Wu Zh.D. Effects of scan rate on the potentiodynamic polarization curve obtained to determine the Tafel slopes and corrosion current density. Corrosion Science. 2009, 51: 581-587.

[DD] McCafferty E. Validation of corrosion rates measured by the Tafel extrapolation method. Corrosion Science 47 (2005) 3202–3215.

  1. Considering Fig.2, at least JCPDS and/or PDF file numbers should be included.
  2. In subsection 3.3, error ranges correlated with microhardness values should be included.
  3. Error bars should be included in Fig. 7
  4. 8 should be revised and, at least, duplicate results should be depicted.
  5. 11 should meticulously be revised and improved. For this purpose, the y axis should be rescaled in “normal” scale. With this, those values indicated in Table 4 should also be revised. This due to “um.cm-2” added with “10-3A/cm2” are completely incompatible or erroneously described.
  6. In Fig. 11, arrows to indicate Tafel extrapolations should be included.
  7. It is hardly suggested that a revision complete should be carried out in order to “rewrite the Captions, subtitles of sections (upper and lower cases).

_ _ _ _

Round 2

Reviewer 1 Report

Still, I do not feel the paper is ready for publication. Besides the errors, the overall structure of the paper is not organized well. Much information was put in the wrong section. Without looking into its technical content, I would recommend against its publication with its current form. Very significant revision is needed. 

  1. “This is also the reason why laser cladding can refine the microstructure of the cladding layer. s reason.” Please correct the sentence.
  2. Abstract: “In order to explore a composite material with both wear resistance and corrosion resistance, Co-Cr alloy coatings with different mass fractions of WC (hard tungsten carbide) hard phases were prepared on Fe316L substrate by laser cladding technology.” First, is the goal of this work to “explore a composite material”? Or should it better be “In order to study the wear resistance and corrosion resistance of a composite material with xxx substrate and xxx coating …”? Second, “(hard tungsten carbide) hard phases” is repetitive. Third, there should be an “a” or “an” before “Fe316L”.
  3. Abstract: “scanning electron microscope (SEM) and energy dispersive spectrometer (EDS)” should be “microscopy” and “spectroscopy”.
  4. Intro: “Based on the research status at domestic and foreign” Still weird. Just say “Based on the current knowledge in the literature”
  5. Materials: “The Fe316L stainless steel plate was selected as the base material sample” please remove “sample”.
  6. Materials: “The Fe316L stainless steel plate was selected as the base material sample, and the base material was polished with 100#, 200#, 300#, and 400# sandpaper respectively, and the oil stains on the surface of the base material were cleaned with acetone solution, and then ultrasonically cleaned with ethanol and dried.” Can authors break it into multiple sentences. This is too long and not readable.
  7. Materials: “The powder has low cost, high hardness, and low temperature requirements for the cladding pool. It is suitable for shaft parts that are easily deformed at high temperature [17]. The overmolded coating has high wear resistance and can greatly improve the degree of friction and wear.” Shouldn’t this information appear in the introduction section? In the intro, the authors should expand the discussion on why the Co-Cr was chosen for this study, which I believe includes these properties mentioned, with other scientific reasons.
  8. Materials: “In order to enhance the hardness, wear resistance and corrosion resistance of Ni-based and Co-Cr-based alloy cladding layers, different mass fractions of WC hard phases were added to enhance the comprehensive mechanical properties of composite powders. WC hard phase has high melting point, high hardness, excellent wear resistance and corrosion resistance, and can strengthen the comprehensive mechanical properties of Ni-WC-based and Co-Cr-WC-based cladding layers. Its physical properties are shown in table 2-2 shown.” Again, this information should appear in the intro section. Did authors do experiments to determine these properties listed in Table 2-2? If not, it should be in the intro section.

Reviewer 3 Report

Some minor/moderate English editing issues need to be fixed.

I can agree with publishing of the manuscript.

Reviewer 4 Report

Please, a minor revision is suggested. Table shou be revised and error ranges included.
